# Identification of Customer Churn Considering Difficult Case Mining

**Jianfeng Li, Xue Bai, Qian Xu *** and **Dexiang Yang**

School of Economics and Management, China Jiliang University, Hangzhou 310018, China;
06b0705071@cjlu.edu.cn (J.L.); 15156205786@163.com (X.B.); s20071201034@cjlu.edu.cn (D.Y.)
* Correspondence: xuqian@cjlu.edu.cn

**Abstract:** In the process of user churn modeling, due to the imbalance between lost users and retained users, the use of traditional classification models often cannot accurately and comprehensively identify users with churn tendency. To address this issue, it is not sufficient to simply increase the misclassification cost of minority class samples in cost-sensitive methods. This paper proposes using the Focal Loss hard example mining technique to add the class weight $\alpha$ and the focus parameter $\gamma$ to the cross-entropy loss function of LightGBM. In addition, it emphasizes the identification of customers at risk of churning and raises the cost of misclassification for minority and difficult-to-classify samples. On the basis of the preceding ideas, the FocalLoss_LightGBM model is proposed, along with random forests, SVM, XGBoost, and LightGBM. Empirical analysis based on a dataset of credit card users publicly available on the Kaggle website. The AUC, TPR, and G-mean index values were superior to the existing model, which can effectively improve the accuracy and stability of potential lost users.

**Keywords:** user churn; unbalanced data; difficult case mining; focal loss function; LightGBM model





## 1. Introduction

In the context of current market conditions and increasing enterprise competition, the differences between products and services continue to diminish. Businesses have gradually shifted their marketing strategies from focusing on products to focusing on customers. The enterprise's primary objective should be to reduce user churn [1]. In terms of product price positioning, marketing strategy, and service enhancement, the development of a new user incurs enormous expenses for an enterprise. Promotion and publicity will convince potential customers more effectively. Therefore, retaining old customers and recognizing the value of old users are essential for expanding business and expanding the market. This has played a significant role in strengthening businesses' competitive advantage in the same industry.

In response to user churn, the historical information of customers is one of the most-valuable assets for businesses and managers. It can be utilized to develop loss-prone customer identification models [2]. Big data analysis technology can be combined with data-mining algorithms to discover the laws contained in historical data, and through the development of mathematical models and other techniques, the data value is converted into reusable, inheritable knowledge [3]. The development of emerging technologies and data-mining technologies has enabled comprehensive research on customer loss forecasts in many industries, including the financial industry. However, big data analysis is still lacking in customer loss forecasting [4]. Applying big data analysis to the enterprise's historical user transaction data, developing an effective user loss model for user behavior analysis, and achieving early warning of users at risk of loss comprise is the crucial method for matching product function and operational strategy.

Fundamentally, user churn identification and early warning are a double classification problem, that is a two-class classification problem; there are only two possibilities of user

loss and user retention. This experiment mainly analyzed the data of user churn. As part of customer churn prediction, Li et al. (2018) utilized the LR, SVM, alternate, and genetic algorithms to address supervised learning and proportional label learning [5]. De et al. (2018) addressed the problem that decision trees and logistic regression models have difficulty handling linear relationships and interactions and proposed using decision rules for customer categorizations [6]. To improve the traditional deep neural network model for UCI public bank employee churn, Mundada et al. (2019) used the Tukey outlier preprocessing method, feature scaling, and the Adam optimization algorithm [7]. Using the user data provided by KKBOX, a music information service, Gregory (2018) processed the time series data using a method of time-sensitive feature engineering [8]. In order to predict user churn, he developed a weighted average model based on XGBoost and LightGBM. According to Wang et al. (2019), the subscriber churn problem in advertising business management can be solved by extracting static and dynamic features from the long-term data of subscribers on advertising platforms and using the GBDT algorithm to predict whether subscribers will churn in the future [9]. Zhang et al. (2014) used the adaptive Boosting algorithm combined with CART regression, used samples to train and test the model, and demonstrated through experiments that the method was applicable in the community setting [10]. Ahmad et al. (2019) assisted telecom operators with predicting subscriber churn by extracting social network features and hybrid sampling of raw data using a combination of the RF, DT, GBM, and XGBoost algorithms [11].

Typically, the number of churned customers is small compared to the number of retained customers. Therefore, it is difficult to identify churned customers using general classification algorithms, as only the majority of class samples (retained users) are accurately identified. Therefore, the classification algorithms used in traditional business processes perform poorly on an overall basis in classifying churned and retained customers [12]. As of today, there are two main levels of solutions to the classification problem for unbalanced data: the processing of data and the improvement of algorithms. Data processing is a method for reducing the imbalance of the original data distribution, with data resampling techniques being the most-prevalent [13,14]. The optimization of standard classification algorithms is the primary focus of algorithm improvement. In existing studies, using cost-sensitive learning to increase the misclassification cost of a few classes of samples and focusing on the classification accuracy of churned customers have been used to identify customer churn. According to Bahnsen et al. (2015), cost-sensitive learning was introduced into the random forest, logistic regression, and decision tree algorithms, and a measurement method was proposed that considered the customer churn cost, which resulted in a 26.4% savings in financial costs on cable supplier data [15]. For the purposes of predicting customer churn in telecommunications, Luo et al. (2010) developed the plain Bayesian, logistic regression, multilayer perceptron, and multilayer perceptron algorithms based on cost-sensitive learning theory [15]. Based on the high-dimensional unbalanced data of telecom customer churn, Özmen et al. (2020) proposed a multi-objective cost-sensitive ant colony optimization algorithm, which minimizes the cost of misclassification while also minimizing the number of features [16]. Wong et al. (2020) introduced cost-sensitive learning into the field of deep learning, proposing a cost-sensitive deep neural network and its ensemble learning version [17]. At the same time, random under-sampling and hierarchical feature extraction were applied to the hidden layer of the deep neural network to improve its generalization ability. In an analysis of user churn prediction, Al-Madi et al. (2018) used Genetic Programming with Cost-Sensitive Learning (GP-CSL) as an optimization algorithm, and the authors concluded that the GP-CSL method was able to identify churned users better in the case of a high penalty cost [18]. By using resampling methods and cost-sensitive learning methods that increase the misclassification cost of churned customer samples, scholars have primarily addressed the issue of imbalance between the number of churned and retained users in the user-churn-modeling process. Using the resampling method, data labels can be balanced by generating samples from a few categories. However, resampling data based only on the information contained in the current few categories

of samples will result in a lack of diversity in the data, as well as generate noise during the sampling process, making it more difficult to differentiate between different types of samples. Although the cost-sensitive method includes a misclassification penalty cost in the training loss of the model, it assigns weights to a relatively small number of samples from a relatively small number of classes, does not distinguish between individual samples within each class, and does not take into account dynamically adjusting attention to different samples based on training results during the model's training process. In the analysis of unbalanced data, dividing the dataset based on the number of samples in each category produces minority versus majority samples; dividing the dataset based on the difficulty of the classifier results in hard versus easy samples. The problem of imbalance between classes and between easy and difficult samples is a significant factor contributing to the lack of sufficient certainty in the classifier to discriminate between classes, which results in the output value being close to the decision threshold [19]. It is, therefore, important to take into account the cost of misclassifying a few classes of samples in the imbalanced data classification problem of identifying user churn, as well as the churned users judged as difficult samples during the training process, so that the user churn model takes into account the deeper mining of difficult cases as well.

This paper incorporated a loss function based on the Focal Loss function that focuses on both minority samples and difficult-to-score samples into the Light Gradient Boosting Machine (LightGBM) classification model, based on the analysis presented above. The addition of category weights and focus parameters to LightGBM's original cross-entropy loss function addresses positive–negative sample imbalance and simple–difficult sample imbalance, respectively, and dynamically adjusts the sample loss contribution during the training process of the model, which results in the user churn of FocalLoss_LightGBM based on difficult case mining. In order to accomplish this, the article analyzed credit card transaction data published on the Kaggle data science website and provided by commercial banks. In addition, the article compared the constructed model to Support Vector Machines (SVMs), Random Forests (RFs), Extreme Gradient Boosting (eXtreme Gradient Boosting (XGBoost)), and the original LightGBM model.

Based on the consideration of the problem of unbalanced data, this experiment focused on a small number of samples and samples that were difficult to score and proposes a user churn FocalLoss_LightGBM model based on difficult case mining, which can effectively identify user churn and has high stability.

## 2. Focal Loss_LightGBM

### 2.1. Cross-Entropy Loss Function for Focal Loss Optimization

2.1.1. Cross-Entropy Loss Function

In the classification problem, the cross-entropy loss function is a type of loss function commonly used in various classification algorithms. To understand cross-entropy, we must first understand information entropy and Kullback–Leibler divergence:

(1)  Information entropy:

Information entropy is a measure of the amount of information required to eliminate uncertainty. The smaller the information entropy, the more certain the information is. Information entropy is used to represent the expectation of all information, which is the probability of each possible outcome in an experiment multiplied by the sum of its outcomes. Let $p(x)$ be the expected output probability distribution of the classifier for sample $x$, expressed as:

$$H(x) = -\sum_{i=1}^{n} p(x) \log(p(x)) \tag{1}$$

The amount of information is represented as:

$$I(x) = -\log(p(x)) \tag{2}$$

(2)    Kullback–Leibler divergence:

The Kullback–Leibler divergence is also called the KL divergence. If there are two separate probability distributions $p(x)$ and $q(x)$ for the same random variable, $p(x)$ is the same as the above statement, and $q(x)$ is the predicted output probability distribution for sample $x$, then the relative entropy can be used to measure the difference between these two probability distributions. The formula is expressed as:

$$D_{KL}(p||q) = \sum_{i=1}^{n} p(x) \log\left(\frac{p(x)}{q(x)}\right) \tag{3}$$

(3)    Cross-entropy:

Cross-entropy is used to measure the approximation of two distributions. The smaller the cross-entropy is, the closer the two probability distributions are. The KL divergence is the difference between the cross-entropy and information entropy, so the cross-entropy of $p(x)$ and $q(x)$ is expressed as:

$$H(p,q) = -\sum_{x} p(x) \log q(x) \tag{4}$$

In the dichotomous classification task, the label of the sample $y_i \in \{0,1\}$, the predicted output $\hat{y}_i \in \{0,1\}$, is used to set a total of $N$ samples. Consequently, the cross-entropy loss function of dichotomous classification is expressed as follows:

$$L(\hat{y}_i, y_i) = -\sum_{i=1}^{N}[y_i \log(\hat{y}_i) + (1 - y_i) \log(1 - \hat{y}_i)] \tag{5}$$

2.1.2. Focal Loss Function

A balance must be achieved between the number of positive and negative samples in order to solve the classification problem of unbalanced data. Weights $\alpha \in (0,1)$ are introduced in order to increase the contribution of a few classes of samples to the training loss of the model, which results in a weighted cross-entropy loss function.

$$L(\hat{y}_i, y_i) = -\sum_{i=1}^{N}[\alpha y_i \log(\hat{y}_i) + (1 - \alpha)(1 - y_i) \log(1 - \hat{y}_i)] \tag{6}$$

Since the weighted cross-entropy loss function can only solve the problem of imbalance between positive and negative samples and although the larger proportion of easy-to-fit samples continues to play a dominant role in the process of increasing model training loss, the classifier will eventually overlearn the easy-to-fit samples [20]. In this paper, the loss function was further optimized using the Focal Loss [21], and a focus parameter was added to the category weight. Therefore, the original cross-entropy loss function is capable of alleviating the problem of reduced model performance caused by the use of too many easy-to-fit samples while focusing on a few classes of samples, which increases the likelihood of identifying lost users.

In the unbalanced binary classification problem, samples with category label 1 are defined as being in the minority class, while samples with category label 0 are considered to be in the majority class. $\hat{y}_i$ represents the probability of determining the category of the ith sample as 1, $1 - \hat{y}_i$, and then, the probability of determining the category as 0. When the true label of the ith $y_i$ sample is 1, $\hat{y}_i$, if it converges to 1, means that the sample is an easy sample $y_i(1 - \hat{y}_i)^\gamma$, and it converges to 0. As a result, the loss contribution of the easy sample is reduced. On the contrary, $\hat{y}_i$, if it converges to 0, means that the sample is a difficult sample, which converges to 1, and the loss contribution of the difficult sample is enhanced. As a result, the difficult case mining problem has been solved. In the

equation, $y_i(1 - \hat{y}_i)^\gamma$ represents the modulation coefficient. Following is an expression for the cross-entropy loss function optimized using Focal Loss:

$$
\begin{aligned}
L(\hat{y}_i, y_i) = -\sum_{i=1}^{N} [\alpha y_i (1 - \hat{y}_i)^\gamma + \\
(1 - \alpha)(1 - y_i)\hat{y}_i^\gamma \log(1 - \hat{y}_i)]
\end{aligned}
\tag{7}
$$

### *2.2. LightGBM Design*

Microsoft's LightGBM [22] is an enhanced model based on the Gradient Boost Decision Tree (GBDT) framework, a lightweight gradient booster [23]. LightGBM employs the histogram algorithm in lieu of the conventional pre-sorting algorithm, as well as the Gradient-based One-Sided Sampling (GOSS) and Exclusive Feature Bundling (EFB) algorithms to enhance operational efficiency [24]. Compared to the GBDT algorithm, the LightGBM algorithm has significantly improved training speed and space efficiency, making it more suitable for training massive and high-dimensional datasets. The primary techniques used by the LightGBM algorithm are described in the following section.

#### 2.2.1. Algorithm for One-Sided Gradient Sampling

During model training, samples with higher gradients contribute more to information gain. To begin, GOSS retains samples with larger gradients, randomly samples them with smaller gradients, and assigns weights to the retained samples with smaller gradients, thus resolving the time-consuming problem of high-dimensional massive data, thereby ensuring that the samples with smaller gradients are given more consideration and the estimation is accurate.

#### 2.2.2. Reciprocal Feature Bundling Algorithm

The EFB algorithm, on the other hand, can combine two mutually exclusive features into a single feature and segment the values of each bundled feature using a histogram algorithm. The basic idea of the histogram algorithm is to discretize continuous features into k discrete features, that is the idea of binning bins, and construct a histogram with a width of k for statistical information (containing k bins). Using the histogram algorithm, we do not need to traverse the data, we only need to traverse k bins to find the best split point, replace the original floating-point value with the discretized value for calculation, count the number of samples in each bin, and find the optimal sharding point according to the discrete value, and the number of slices that need to be divided is equal to the number of bins minus one.

As depicted in Figure 1, the histogram algorithm discretizes the continuous feature values into integers while simultaneously constructing the histogram. After a single pass through the data, the histogram algorithm can be used to store large-scale data in the histogram and determine the optimal partition point based on the values after feature discretization. The statistical process of the histogram algorithm is depicted in Figure 1.

#### 2.2.3. Grow-by-Leaf Strategy with Depth

In contrast to the levelwise growth strategy adopted by the GBDT algorithm, LightGBM employs the leafwise growth Strategy with a depth limit, i.e., finds the leaf node with the highest splitting gain. As illustrated in Figure 2, the leafwise growth strategy splits all leaf nodes in the same layer simultaneously, which results in unnecessary splitting and searching on a large number of leaf nodes with low information gain. LightGBM selects the grow-by-leaf strategy with depth restriction to select the leaf nodes with the highest information gain for splitting and then sets the maximum depth of the decision tree to prevent overfitting. The grow-by-layer and grow-by-leaf strategies are depicted in Figures 2 and 3, respectively.

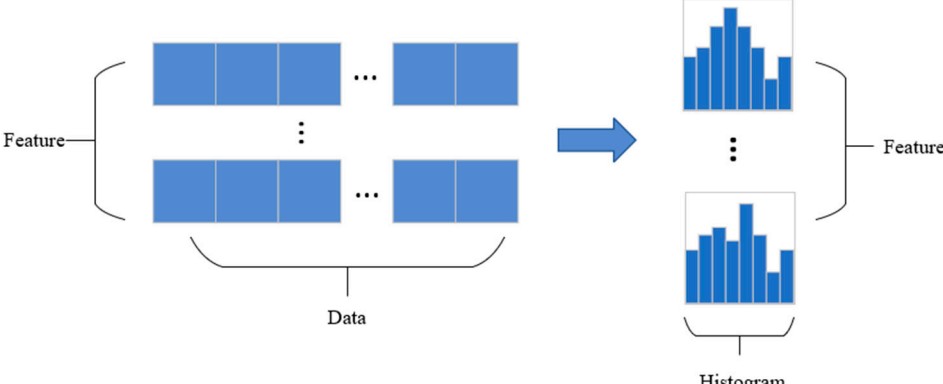

**Figure 1.** Histogram algorithm.

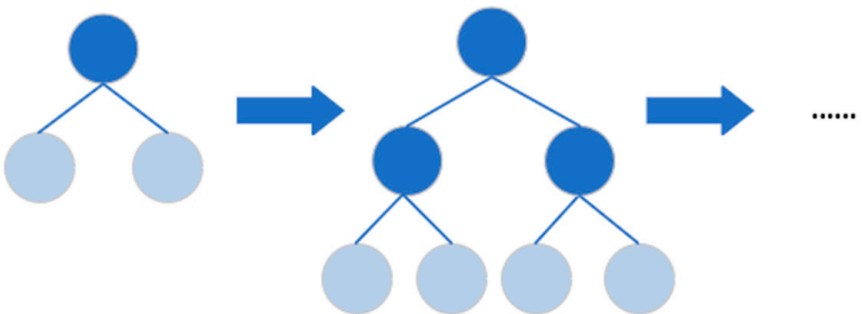

**Figure 2.** Growth strategy by layer.

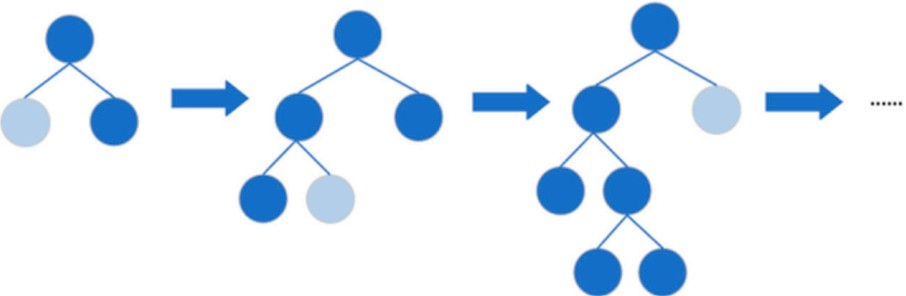

**Figure 3.** Growth strategy by leaves.

## 3. Empirical Analysis

### 3.1. Data Preparation and Problem Description

The dataset utilized in this paper was the publicly accessible Credit Card Customers dataset on Kaggle. For a total of 10,127 credit card users with 23 indicators for each user, the data include personal information such as age, gender, number of dependents, and educational status, as well as account information, including credit limit, total working balance, and transaction amount changes.

Figure 4 displays sample data from 10,127 credit card users collected from a bank during the course of its credit card business. It illustrates that the average age of credit card users is between 30 and 60 years old, with a normal distribution. Secondly, the total sample of users collected contains an equal number of males and females, and the majority of users have relatives or friends who require assistance. The highest percentage of credit card users possess a Bachelor's degree, and the majority of credit card holders fall into the low- and middle-income brackets.

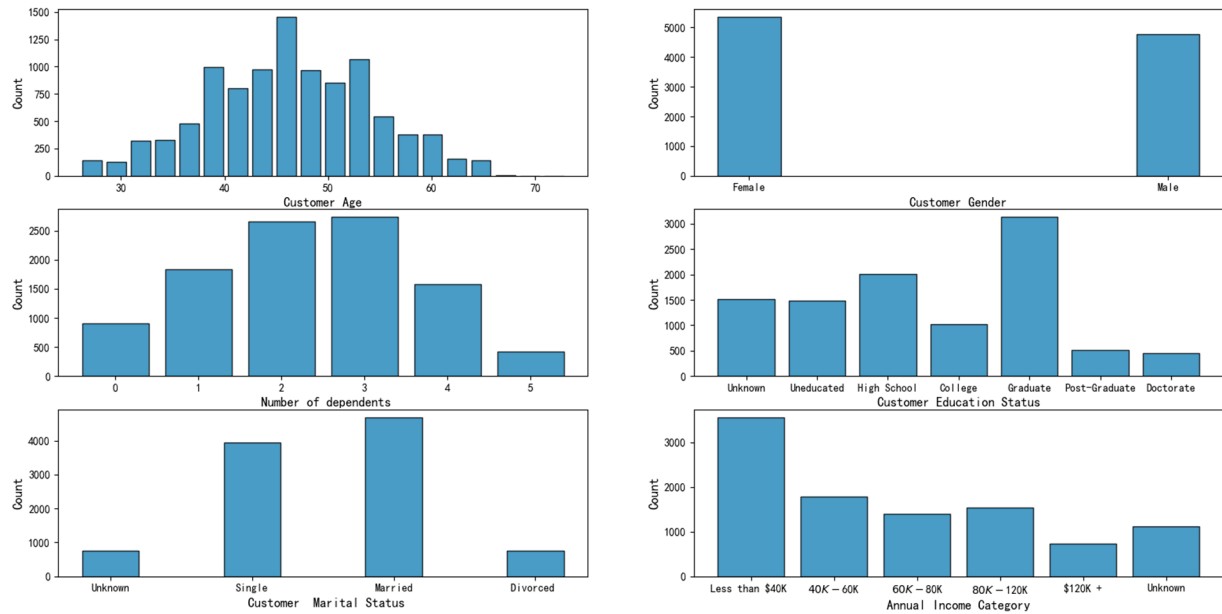

**Figure 4.** Descriptive statistics of user information variables.

The remaining 17 variables describe primarily information related to customers' credit card transactions, etc. Tables 1 and 2 describe the credit card type variables and numeric variables.

**Table 1.** Descriptive statistics of type variables.

| Credit Card Color | Number of Users |
| --- | --- |
| Blue | 9436 |
| Silver | 555 |
| Gold | 116 |
| Platinum | 20 |

**Table 2.** Descriptive account information.

| Customer Bank Transaction Status Variable | Minimum Value | Maximum Value | Mean Value | Standard Deviation |
| --- | --- | --- | --- | --- |
| Time to establish relationship with bank | 13 | 56 | 35.9284 | 7.9864 |
| Total number of products owned by customers | 1 | 6 | 3.8126 | 1.5544 |
| Inactive months in the last 12 months | 0 | 6 | 2.3412 | 1.0106 |
| Number of contacts in the last 12 months | 0 | 6 | 2.4553 | 1.1062 |
| Credit card line of credit | 1438.3 | 34,516 | 8631.9537 | 9088.7767 |
| Total credit card revolving balance | 0 | 2517 | 1162.8141 | 814.9873 |
| Open purchase credit line (past average of 12 months) | 3 | 34,516 | 7469.1396 | 9090.6853 |
| Change in transaction amount (4th quarter over 1st quarter) | 0 | 3.397 | 0.7599 | 0.2192 |
| Total transaction amount (last 12 months) | 510 | 18,484 | 4404.0863 | 3397.1293 |
| Total number of transactions (last 12 months) | 10 | 139 | 64.8587 | 23.4726 |
| Change in transaction count (4th quarter) over 1st quarter | 0 | 3.714 | 0.7122 | 0.2381 |
| Average card utilization | 0 | 0.999 | 0.2749 | 0.2757 |

There are 8500 retained customers and 1627 churned customers in the aforementioned credit card customer data, indicating an obvious imbalance. Identifying churned customers is difficult due to the imbalance of the data, and the standard classification trained by the general machine learning algorithm tends to be biased toward the majority of samples and less accurate for the smaller proportion of churned customers. Consequently, methods that

accurately identify customers at risk of churn are required. The distribution of customer type variables is shown in Table 3.

**Table 3.** Distribution of variables regarding customer types.

| Customer Type | Number | Percentage |
|---|---|---|
| Churned customers | 1627 | 16.07 |
| Retained customers | 8500 | 83.93 |

### 3.2. Model Comparison Analysis and Evaluation

First, this paper divided the aforementioned dataset of credit card users. This paper performed 30 divisions on the original dataset with a 3:7 ratio of the test set to the training set in order to prevent the randomness of the data division from affecting the model's performance. This allowed the model to be trained on 30 distinct training sets. Second, the FocalLoss_LightGBM model and the SVM, random forest, XGBoost, and unimproved LightGBM models, which are comparative user attrition models, were obtained through 30 repetitions on the divided training dataset, and the average value was calculated to assess the model accuracy by comparing the values from each experiment individually. In order to compare the stability of each user churn identification model, on 30 experiments, we calculated the standard deviation of the FocalLoss_LightGBM user churn model and each comparison model. Figure 5 depicts the general process of constructing the FocalLoss_LightGBM model for user churn based on hard case mining.

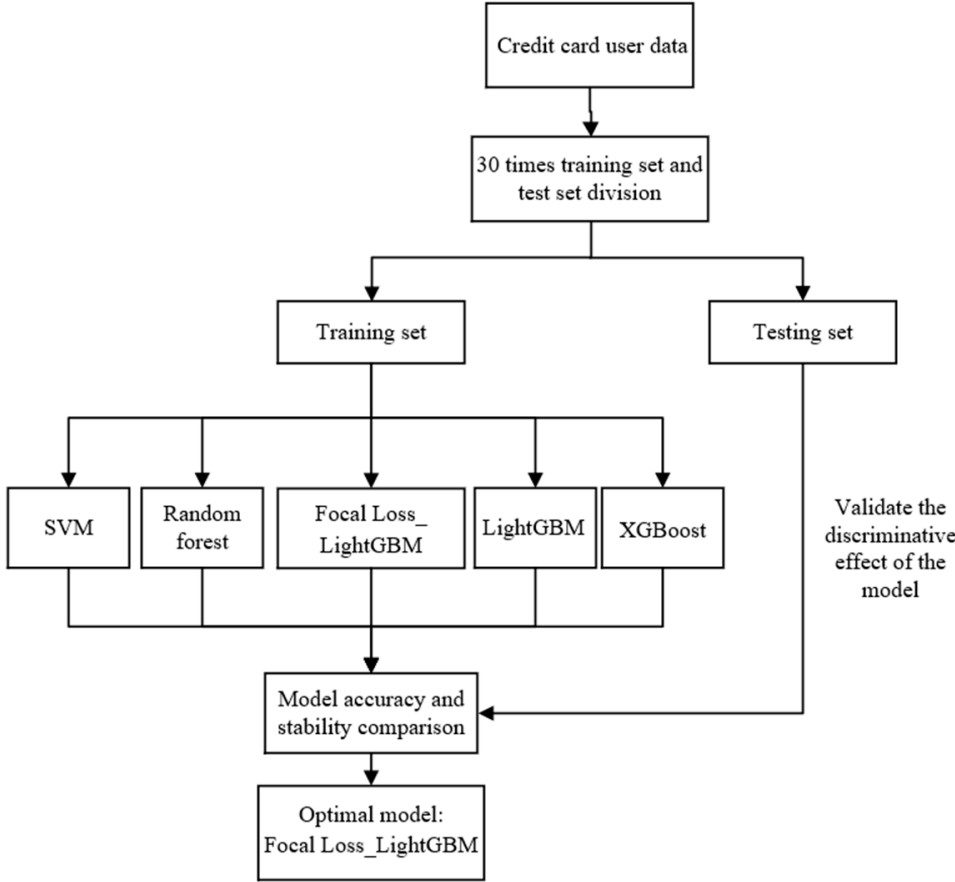

**Figure 5.** The FocalLoss_LightGBM model.

### 3.2.1. Introduction of Evaluation Metrics

Due to the fact that user churn identification involves a binary classification problem with unbalanced data, the classification accuracy index alone cannot accurately reflect the

effect of the model in identifying potentially churned users. The FocalLoss_LightGBM user churn model constructed in this paper was evaluated using four indicators: TPR, AUC, f1-score, and G-mean. This paper combined the actual requirements for identifying user churn:

(1)  True Positive Churn Identification Rate (TPR):

The TPR represents the classification rate of all churned customer samples. Improving the classification rate of churned customers was a key objective of this study. The TPR represents the classification rate of all churned customers, which reflects the model's ability to identify churned customers. The TPR is calculated as follows. The TPR is also known as the recall rate. The calculation method is expressed as follows:

$$\text{TPR} = \frac{\text{TP}}{\text{TP} + \text{FN}} \tag{8}$$

(2)  AUC:

The Area Under the ROC Curve (AUC) is an evaluation metric that measures the degree of merit of a binary classification model. It indicates the probability that a positive example sample will rank higher than a negative example sample, allowing a more comprehensive evaluation of classification performance on unbalanced samples [25,26]. The AUC is the area under the Receiver Operating Characteristics (ROC) curve, which is a curve plotted with the False Positive Rate (FPR) value as the horizontal axis and the True Positive Rate (TPR) value as the vertical axis. The FPR calculates the proportion of retained users who are incorrectly classified as churned users. The calculation method is expressed as follows:

$$\text{FPR} = \frac{\text{FP}}{\text{FP} + \text{TN}} \tag{9}$$

(3)  f1-score:

The f1-score is the average of the Precision (P) and recall (TPR), which is a balanced combination of the two. The calculation method is expressed as follows:

$$\text{P} = \frac{\text{TP}}{\text{TP} + \text{FP}} \tag{10}$$

$$\text{f1\_score} = \frac{2\text{P} \cdot \text{TPR}}{\text{P} + \text{TPR}} \tag{11}$$

(4)  G-mean

The G-mean combines the classification accuracy of two sample types, and the G-mean is sufficient to measure the classification effectiveness of classification methods on unbalanced datasets in comparison to the overall correct classification ratio [27]. The G-mean is the geometric mean of the correct classification ratios in the samples from the minority and majority classes.

$$\text{G\_mean} = \sqrt{\frac{\text{TP}}{\text{TP} + \text{FN}} \times \frac{\text{TN}}{\text{FP} + \text{TN}}} \tag{12}$$

3.2.2. Comparative Analysis of User Retention Model Precision

The parameters were adjusted using five-fold cross-validation. Thirty trials were conducted on each unique training dataset, and the mean values of each metric were calculated for comparison. Tables 4 and 5 present the parameter definitions and optimal settings for each classification model utilized in this study.

**Table 4.** Meaning and setting of parameters of SVM, random forest, and XGBoost models.

| Identification Model | Parameter | Parameter Meaning | Optimal Setting Value |
|---|---|---|---|
| SVM | C | Penalty factor | 10,000 |
|  | kernel | Kernel function type | "rbf" |
|  | Gamma | Kernel function coefficient | 0.001 |
| Random forest | n_estimators | Number of decision trees | 31 |
|  | max_depth | Maximum depth of the tree | 11 |
|  | min_samples_split | Minimum number of samples needed to split internal nodes | 50 |
| XGBoost | n_estimators | Number of decision trees | 300 |
|  | max_depth | Maximum depth of the tree | 4 |
|  | learning_rate | Learning rate | 0.2 |
|  | booster | Weak learner type | "gbtree" |

**Table 5.** Meaning and setting of parameters of LightGBM and FocalLoss_LightGBM models.

| Main Parameters of the LightGBM Model and the FocalLoss_LightGBM Model | Parameter Meaning | LightGBM | FocalLoss_ LightGBM |
|---|---|---|---|
| num_boost_round | Maximum number of iterations | 162 | 293 |
| learningrate | Learning rate | 0.1 | 0.1 |
| max_depth | Maximum depth of the tree | 7 | 5 |
| num_leaves | Number of leaves | 65 | 10 |
| feature_fraction | Feature random sampling ratio | 0.8 | 0.9 |
| bagging_fraction | Sample random sampling ratio | 0.6 | 0.9 |
| $\alpha$ | Category weight |  | 0.95 |
| $\gamma$ | Focusing parameter |  | 0.1 |

Using the grid tuning results, each recognition model was set, and 30 trials were conducted on different training sets. The mean values of the predicted output indicators were calculated for each experiment, and the results are displayed in Table 6. Both SVM and random forest produced similar scores, but their classification accuracy for small samples was around 0.7, indicating that only 70% of churned users were correctly identified and that the recognition of churned users was inadequate. It was demonstrated that the XGBoost and LightGBM algorithms significantly improved both the correct identification rate of churn, as well as the accuracy of the classification of retained customers when compared to the first two algorithms. Specifically, the LightGBM algorithm achieved a mean AUC of 0.99 across 30 trials, indicating that this method takes into account the classification accuracy of both minority and majority class samples and does not improve the churn recognition rate at the expense of a lower correct classification rate for retained users. By comparing the FocalLoss_LightGBM model constructed in this paper to other methods, it is possible to conclude that using the Focal Loss method to optimize the original cross-entropy loss function did not just inherit the benefits of the LightGBM model in terms of overall credit card user identification accuracy, but also enhanced the identification of churned users. There was an increase in the churn rate from 0.8691 to 0.9418.

**Table 6.** Comparison of user churn model accuracy evaluation results.

| Churn Model | Average Value of AUC | TPR Mean Value | Mean Value of f1-Score | G-Mean |
|---|---|---|---|---|
| SVM | 0.8461 | 0.7266 | 0.7616 | 0.8376 |
| RF | 0.8449 | 0.7038 | 0.7915 | 0.8329 |
| XGBoost | 0.9344 | 0.8793 | 0.9088 | 0.9327 |
| LightGBM | 0.9910 | 0.8691 | 0.8954 | 0.9258 |
| FocalLoss_LightGBM | 0.9937 | 0.9418 | 0.9045 | 0.9573 |

Consequently, the FocalLoss_LightGBM user churn model was significantly more accurate than the other four models, and it can be applied to actual churn identification business scenarios, allowing enterprises to tailor customer retention strategies, as well as implement marketing strategies more accurately. Furthermore, the proposed model takes into account the needs of the customer. In addition, the proposed model is capable of identifying both churned and retained users and does not incorrectly classify retained users as churned users at the cost of improving the churn identification accuracy, which is of practical importance for enterprises seeking to reduce the cost of user relationship management.

### 3.2.3. User Churn Analysis of Model Stability

To make the churn recognition model more generalizable to a broader range of business scenarios, the stability of the churn recognition model's feasibility must be evaluated to determine whether or not the model has strong generalization capability. In this section, we calculated the standard deviation of the evaluation indexes for the FocalLoss_LightGBM churn model and the other four comparative models in 30 experiments to determine the dispersion of each model for churn identification, as well as facilitate an intuitive comparison of the user churn model with the other models. After 30 experiments, Table 7 displays the maximum value, minimum value, and standard deviation of the evaluation indices for the five models.

**Table 7.** Comparison of user churn identification stability evaluation results.

| Model | AUC | | | TPR | | | f1-Score | | | G-Mean | | |
|---|---|---|---|---|---|---|---|---|---|---|---|---|
| | Max | Min | Std | Max | Min | Std | Max | Min | Std | Max | Min | Std |
| SVM | 0.8679 | 0.8272 | 0.0091 | 0.7728 | 0.6908 | 0.0185 | 0.7846 | 0.7385 | 0.0131 | 0.8627 | 0.8159 | 0.0105 |
| RF | 0.8758 | 0.8232 | 0.0138 | 0.7639 | 0.6618 | 0.0274 | 0.8337 | 0.7564 | 0.0191 | 0.8686 | 0.8079 | 0.0162 |
| XGBoost | 0.9461 | 0.9206 | 0.0067 | 0.9059 | 0.8515 | 0.0138 | 0.9220 | 0.8949 | 0.0081 | 0.9452 | 0.9180 | 0.0071 |
| LightGBM | 0.9944 | 0.9864 | 0.0020 | 0.9039 | 0.8396 | 0.0140 | 0.9146 | 0.8812 | 0.0081 | 0.9373 | 0.9137 | 0.0073 |
| FocalLoss_ LightGBM | 0.9956 | 0.9914 | 0.0011 | 0.9683 | 0.9159 | 0.0119 | 0.9183 | 0.8920 | 0.0077 | 0.9675 | 0.9469 | 0.0046 |

The results of the AUC, churned user recognition correct rate, TPR, f1-score, and G-mean on five recognition models for 30 experiments are depicted in Figures 6–9. Based on these four figures, it was evident that the SVM and random forest models were incapable of identifying churned users and that the recognition accuracy of both models was unstable, which resulted in poor model stability. The LightGBM and XGBoost models had significantly higher recognition accuracy for churned users than the first two, and based on the numerical fluctuations of each index, their recognition performance was more stable, as shown in Table 7 and Figures 6–9. Lastly, based on the comparison between the FocalLoss_LightGBM user churn model and other models in each figure, the proposed method not only outperformed the other models in terms of churn recognition accuracy, but it also demonstrated greater stability in recognition performance.

Therefore, based on the above comparison and analysis of the accuracy and stability of each user churn model, the FocalLoss_LightGBM model proposed in this paper not only accurately identified credit card users with a tendency to churn, but also maintained high levels of stability across multiple experiments. The model can be applied to other industries with unbalanced user categories in order to anticipate churned customers in advance and implement retention policies that are specifically tailored to them. The model can be applied to other industries to accurately identify churn.

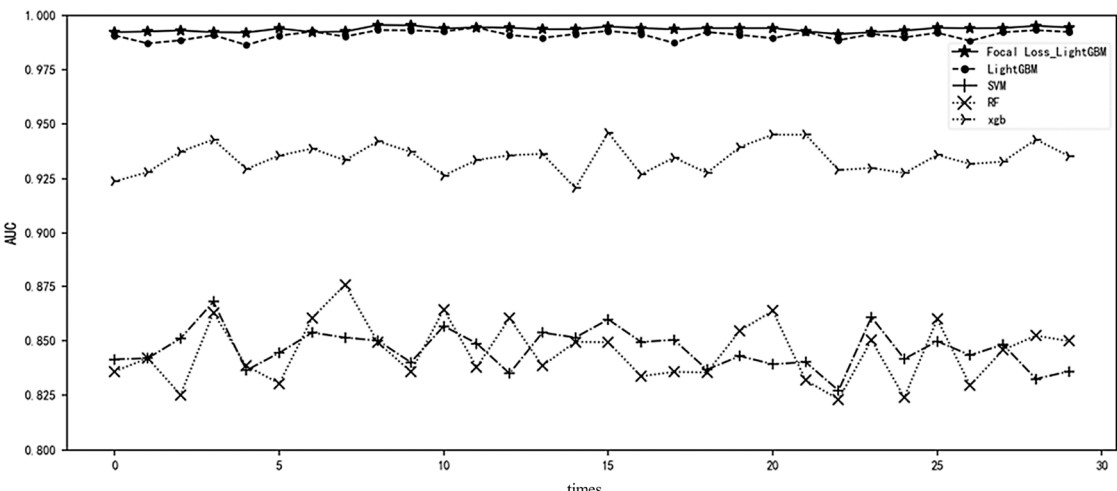

**Figure 6.** AUC index 30-times test comparison.

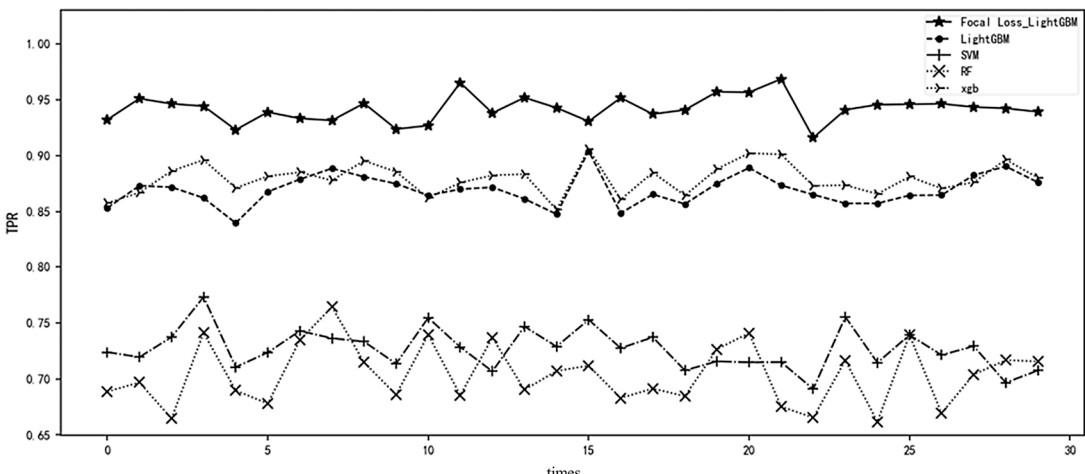

**Figure 7.** TPR index 30-times test comparison.

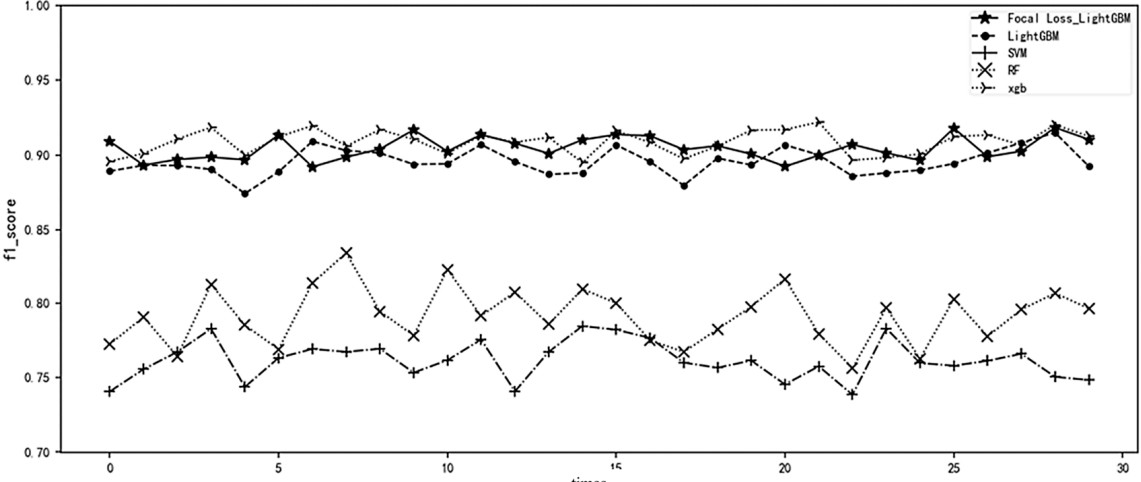

**Figure 8.** f1_socre index 30-times test comparison.

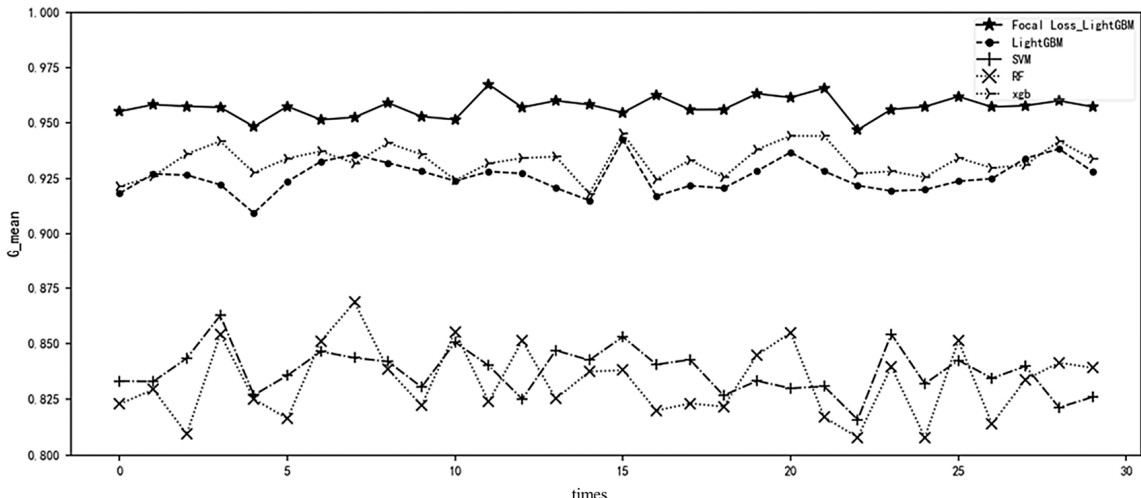

**Figure 9.** G-mean indicator 30-times test comparison.

## 4. Conclusions

The proportion of churned customers is low in the study of user churn, which results in the unbalanced nature of historical user data. Because of this, it is difficult to improve the model's accuracy in identifying potential churned users using general machine learning models and a single prediction accuracy metric. This paper optimized the original cross-entropy loss function and introduced category weights and focus parameters to control the weights of positive and negative samples, as well as simple–difficult samples and adjusted the misclassification cost of the samples based on the proportion of samples and their classification difficulty in each training round to construct a user churn model based on difficult cases, FocalLoss_LightGBM. The results demonstrated that, in comparison to support vector machine, random forest, and LightGBM, the proposed model not only identified churned users with greater precision, but also with greater identification stability across different dataset subsets. The proposed user churn model expands the study of big data analytics for the purpose of identifying potential churned users. Applying the model to the actual user management process can help businesses effectively identify customers with churning propensities, obtain user dynamics, rapidly develop marketing strategies and retention plans, and reduce user relationship management expenses. In addition, as a result of the model's high identification stability across multiple datasets, it can be extended to identify churn in the telecommunications, Internet, and new media industries in future research, and it has a strong application in classification problems involving typical imbalance characteristics, such as the detection of financial fraud default. To filter out the factors that are most-important to user retention from the enormous amount of user data, we will consider incorporating feature extraction of user history data in the future, in order to assist businesses in identifying lost customers and developing more targeted and differentiated strategies for improving customer service.

**Author Contributions:** Methodology, J.L.; software, J.L. and Q.X.; validation, X.B. and D.Y.; formal analysis, J.L.; investigation, X.B.; resources, Q.X.; data curation, D.Y. All authors have read and agreed to the published version of the manuscript.

**Funding:** This paper was supported by the Zhejiang Province Soft Science Research Program Project (2023C35028).

**Data Availability Statement:** Not applicable.

**Conflicts of Interest:** No potential conflict of interest is reported by the authors.

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
