# Peer review of "Identification of Customer Churn Considering Difficult Case Mining"

_systems, doi:10.3390/systems11070325_

Round 1
Reviewer 1 Report
Thank you for the interesting paper.
The paper represents an original research on the identification of the churned customers.
The paper is logically structured, however, the aim of the paper is not clearly stated, therefore it is difficult to assess the contribution of the paper.
Moreover, given the low proportion of churned customers, it is unclear if general conclusions can be drawn.
Author Response
Dear reviewer,
Thank you for giving us the opportunity to modify the paper! Thank you for your helpful suggestions, it is very enlightening for our research.
Although this article is a study on customer churn prediction based on imbalanced samples, it focuses more on minority class samples and difficult to distinguish samples when constructing user churn models. That is, when constructing user churn models, we consider in-depth mining of difficult examples. Although the number of difficult examples of user churn is relatively small and not general, our experiment is based on general research on customer churn prediction based on imbalanced samples, considering special situations, and is more comprehensive.
Reviewer 2 Report
Report for "Identification of Customer Churn Considering Difficult Case Mining" submitted to Systems
The aim of the manuscript is to introduce the Focal Loss Hard Example Mining technique to add class weight and focus parameters to LightGBM. The proposed methodology used to predict customer churn and the analysis are done on a very well-known credit card data set from Kaggle. Please consider the following:
In the introduction, please consider the study that also uses the same data set to predict churn by SVM optimized via Bayesian optimization
Also, please consider the following recent study, which employs ensemble learning on churn analysis: https://doi.org/10.3390/su15118631
At the end of the introduction, please write a brief paragraph that emphasizes the novelty of the study and lists the chapters that follow.
Section 2 is fine.
Do you have case studies for different parameter settings? If you have, could you please discuss and comment on them? Please enrich the conclusion part with the limitations of the algorithm and future study ideas.
Author Response
Dear reviewer,
Thank you for giving us the opportunity to modify the paper! Thank you for your helpful suggestions, it is very enlightening for our research. Based on reviewer’s suggestion and request, we have made corrected modifications on the revised manuscript. The comments was summarized into the following aspects and we will answer them one by one.
1、Report for "Identification of Customer Churn Considering Difficult Case Mining" submitted to Systems。The aim of the manuscript is to introduce the Focal Loss Hard Example Mining technique to add class weight and focus parameters to LightGBM. The proposed methodology used to predict customer churn and the analysis are done on a very well-known credit card data set from Kaggle. Please consider the following:In the introduction, please consider the study that also uses the same data set to predict churn by SVM optimized via Bayesian optimization。Also, please consider the following recent study, which employs ensemble learning on churn analysis: https://doi.org/10.3390/su15118631。At the end of the introduction, please write a brief paragraph that emphasizes the novelty of the study and lists the chapters that follow.
The author’s answer:
Based on the suggestions of the reviewers, we carefully reviewed the paper - Predictive Chun Modeling for Sustainable Business in the Telecommunications Industry: Optimized Weighted Ensemble Machine Learning. This article first uses KNN, GatBoost and Random forest algorithms to predict customer churn, and then uses Powell optimization algorithm to optimize the fusion weight of the three algorithms. The above optimal weighted Ensemble learning model is generally applicable to the unbalanced data problem of user churn. This paper, on the basis of considering the unbalanced data problem, designs a Loss function (Focal Loss function) that focuses on a small number of samples and hard to distinguish samples at the same time, and embeds it into the classification model of the Light Gradient Boosting Machine (LightGBM). Specifically, add category weights to the original Cross entropy Loss function of LightGBM α And focusing parameters γ, We have dealt with the imbalance of positive and negative samples and the imbalance of simple hard to distinguish samples separately, and can dynamically adjust the loss contribution of samples during the training process of the model, ultimately obtaining the FocalLoss of user churn based on hard case mining_ LightGBM model. Compared to the suggested paper, this article is more targeted and specific. Of course, the above paper also has great reference significance. In the future, we can consider using the optimal weighted Ensemble learning model to solve the problem of unbalanced data of user churn based on hard case mining.
Based on the reviewer's suggestions for the last part of the introduction, we have made modifications and summarized the novelty of our algorithm model for addressing the issue of customer churn in imbalanced data.
2、Do you have case studies for different parameter settings? If you have, could you please discuss and comment on them? Please enrich the conclusion part with the limitations of the algorithm and future study ideas.
The author’s answer:
This article did not conduct a case comparison analysis of other parameter algorithm models, but instead used grid search to traverse all possibilities of parameters. Therefore, algorithms with different parameters were run, and the optimal parameters were selected for case comparison analysis, which is representative.
Reviewer 3 Report
In the abstract (first sentence), it is not clear what the authors mean by customer resources. One can only guess. So, please reformulate. The same holds for the second sentence. Although it is clear that it is about customer churn, it should be specified what kind of classification models the authors mean.
Instead of simply suggesting an improved algorithm validate on the basis of one example, the authors shoudl explain in more detail the core rationale of their approach. Otherwise the introduced modification appears to be random.
Line 47-49 should be explained more explictly.
Sect. 2 requires references to the functions used and more explanations.
Figure 1 is difficult to interpret and requires more explanation.
It seems to be mainly fine.
Author Response
Dear reviewer
Thank you for giving us the opportunity to modify the paper! Thank you for your helpful suggestions, it is very enlightening for our research. Based on reviewer’s suggestion and request, we have made corrected modifications on the revised manuscript. The comments was summarized into the following aspects and we will answer them.
- In the abstract (first sentence), it is not clear what the authors mean by customer resources. One can only guess. So, please reformulate. The same holds for the second sentence.
The author’s answer:
Based on the suggestions of the reviewers, we have made modifications to the first two sentences of the abstract and provided a concise statement of the reasons and necessity for conducting this experiment.
- Although it is clear that it is about customer churn, it should be specified what kind of classification models the authors mean. Instead of simply suggesting an improved algorithm validate on the basis of one example, the authors should explain in more detail the core rationale of their approach. Otherwise the introduced modification appears to be random.
The author’s answer:
The classification model in this paper is the Cross entropy Loss function of LightGBM, and innovatively introduces the category weight α And focusing parameters γ, Improving the misclassification cost of minority and difficult to distinguish samples, with a focus on strengthening the identification of customers at risk of churn, as proposed in the fifth and sixth lines of the preface of the paper, and more detailed sections are presented in Chapter 2.1.2 Focal Loss Function Loss section. We have provided a more detailed explanation of the definition of user churn identification and warning in response to the issue that needs to be explained more clearly in the third paragraph of the introduction. We have provided a more detailed explanation of the core idea of the histogram in response to the issue in Figure 1, in order to facilitate its understanding.
- Line 47-49 should be explained more explictly.
The author’s answer:
Thank you for your suggestion. We have revised lines 47 to 49, as shown in the revised manuscript.
- 2 requires references to the functions used and more explanations.
The author’s answer:
We have provided a more detailed explanation of the formula in section 2.1.1 of the article for readers to understand. As shown on page 4-5.
- Figure 1 is difficult to interpret and requires more explanation.
The author’s answer:
We have provided a more detailed explanation of the core idea of the histogram in response to the issue in Figure 1, in order to facilitate its understanding. As shown on page 7.
Round 2
Reviewer 3 Report
The issues are resolved to an acceptable extent.